# Alleviating Effects of Ovatodiolide and Antcin K Supplements on High-Fat Diet-Induced Cardiovascular Dysfunction in ApoE-Knockout Mice by Attenuating Oxidative Stress

**DOI:** 10.3390/nu15184074

**Published:** 2023-09-20

**Authors:** Chen-Wen Lu, Wen-Jhen Wu, Thi Kim Ngan Nguyen, Szu-Chuan Shen, Yeh-B. Wu, Hui-Ju Liang, Chung-Hsin Wu

**Affiliations:** 1School of Life Science, National Taiwan Normal University, Taipei City 11677, Taiwan; kumo.lu@gmail.com (C.-W.L.); efgy78@gmail.com (W.-J.W.); marynguyen@ntnu.edu.tw (T.K.N.N.); scs@ntnu.edu.tw (S.-C.S.); 2ARJIL Pharmaceuticals LLC, Hsinchu City 30013, Taiwan; ybw333@arjilbio.com (Y.-B.W.); kathy@arjilbio.com (H.-J.L.)

**Keywords:** hyperlipidemia, atherosclerosis, cocktail therapy, oxidative stress, inflammation, high-fat diet, ApoE-knockout mice

## Abstract

A high-fat diet (HFD) is a major risk factor for cardiovascular diseases. Many pure compounds have been demonstrated to be effective in treating cardiovascular diseases. In this study, we investigated the alleviating effects of oral ovatodiolide and antcin K (OAK) supplements on HFD-induced cardiovascular dysfunction in apolipoprotein E (ApoE)-knockout mice. Cardiovascular dysfunction was induced in ApoE-knockout mice by feeding them an HFD for 12 weeks. The degree of cardiovascular dysfunction was assessed through echocardiography, hematological and biochemical analyses, and immunofluorescence and immunohistochemical staining. The HFD-fed mice exhibited cardiovascular dysfunction—abnormal blood biochemical index. The arterial wall tissue exhibited the marked deposition of lipids, upregulated expression of vascular cell adhesion molecule-1 and CD36 receptors, and downregulated expression of the ABCA1 receptor. Macrophages isolated from the peritoneal cavity of the mice exhibited increased levels of lipid accumulation, reactive oxygen species, and CD11b expression but reduced mitochondrial membrane potential. The expression of superoxide dismutase 2 was downregulated and that of tumor necrosis factor-α was upregulated in the myocardial tissue. Oral OAK supplements twice a day for 12 weeks significantly mitigated HFD-induced cardiovascular dysfunction in the experimental mice. Oral OAK supplements appear to be a promising strategy for treating HFD-induced cardiovascular dysfunction. The underlying mechanisms may involve the reduction of lipid accumulation in the artery and oxidative stress and inflammation in the cardiovascular tissue.

## 1. Introduction

High-fat diet (HFD)-induced cardiovascular disease is a common cause of mortality in humans [1]. Atherosclerosis is the leading cause of cardiovascular disease [2]. Elevated circulating low-density lipoprotein (LDL) levels, vascular reactive oxygen species (ROS) production, and inflammation are key risk factors for atherosclerosis [3]. Inflammation plays a crucial role in the progression of cardiovascular disease and atherosclerotic lesions [4]. Myocardial infarction resulting from the partial or complete occlusion of coronary arteries due to atherosclerosis and cholesterol buildup often causes cardiovascular morbidity and mortality. This blocks the supply of oxygen and nutrients to the myocardium, eventually leading to myocardial cell death [5]. Hyperlipidemia, a pathological state characterized by elevated serum levels of cholesterol and triglyceride, is a major risk factor for cardiovascular disease. Most patients with myocardial infarction have hyperlipidemia [6]. Dyslipidemia, diabetes, hypertension, and postmetabolic disorders are closely associated with myocardial infarction [7]. The chronic elevation of blood cholesterol levels leads to the development of atherosclerosis and negatively affects the myocardium through increased oxidative stress, mitochondrial dysfunction, and apoptosis. Hypercholesterolemia can cause microvascular dysfunction by inducing oxidative stress and inflammation. These mechanisms may increase the susceptibility of the myocardium to infarction [8].

Certain herbal extracts, such as *Anisomeles indica* and *Antrodia camphorata* extracts, can combat atherosclerosis by producing antibodies or inhibitory molecules that target proatherosclerotic biomarkers [9,10]. Although these herbal extracts have considerable therapeutic potential for atherosclerosis, most of them are yet to be explored for drug development. Ovatodiolide (C_20_H_24_O_4_) is a diterpenoid derivative isolated from the traditional Chinese medicinal herb *A. indica* [10]. This compound is commonly and traditionally used to treat inflammation-associated diseases and thus is a potential immunotherapeutic agent [11,12]. Furthermore, ovatodiolide is effective against inflammation as well as metabolic and immune dysregulation in myocardial infarction [13]. Antcin K (C_29_H_44_O_6_), a triterpenoid derived from the fruiting bodies of *A. camphorate*, exerted inhibitory effects on proinflammatory cytokines in rheumatoid synovial fibroblasts and mitigated hyperlipidemia [14,15]. Thus, ovatodiolide and antcin K (OAK) supplements may have therapeutic potential for cardiovascular dysfunction and atherosclerosis; however, the mechanisms underlying their effects are yet to be completely understood.

In this study, we investigated the effectiveness of a combination therapy involving two pure compounds of OAK supplements in mitigating HFD-induced cardiovascular dysfunction in mice lacking apolipoprotein E (ApoE). In addition, we explored the molecular mechanisms underlying the positive effects of oral OAK supplements on cardiovascular dysfunction. Knocking out ApoE in mice with HFD-induced cardiovascular dysfunction can accelerate the progression of atherosclerosis. This animal model has been widely used to study cardiovascular dysfunction because atherosclerotic lesions spontaneously develop in ApoE-knockout mice [16,17].

## 2. Materials and Methods

### 2.1. OAK Supplements Preparation and High-Performance Liquid Chromatography

OAK supplements comprise two pure compounds, namely, ovatodiolide (Figure 1A) and antcin K (Figure 1B), in specific ratios. For this study, we obtained OAK supplements from ARJIL Pharmaceuticals LLC, Hsinchu, Taiwan. Methods used for OAK supplements have been described previously [18,19]. Dried *A. indica* or *A. camphorata* fruiting bodies were separately ground into fine powders and extracted with 95% ethanol at ambient temperature. The slurry of *A. indica* or *A. camphorata* was filtered, and the filtrate was concentrated under reduced pressure to obtain a crude extract. OAK supplements were separately extracted from the corresponding plants by using hexane and ether, and the extracted samples were separated using silica gel columns with ethyl acetate and hexane. Furthermore, the compounds were purified to >90% purity. The chromatographic fingerprint of OAK supplements was determined through high-performance liquid chromatography (HPLC; Thermo Fisher Scientific Inc., Waltham, MA, USA; Figure 1). The samples were graded with acetonitrile and methanol (Burdick & Jackson Korea, Seoul, Republic of Korea) and qualitatively analyzed within 70 min under specific HPLC conditions. OAK supplements were dissolved in purified water obtained from the Milli-Q water purification system (Millipore, Burlington, MA, USA).

The HPLC conditions were as follows: column (Waters/Sunfire RP18, Milford, MA, USA), 5 µm and 150 mm × 4.6 mm ID; mobile phase, 0.05% trifluoroacetic acid aqueous solution (solution A) and acetonitrile (solution B); detection, photo diode array (λ = 220 nm); injection volume, 10 µL; analytic concentration, 0.4 mg/mL; oven temperature, 30 °C; and run time, 20 min.

### 2.2. DPPH Assay

We performed the 1,1-diphenyl-2-picrylhydrazine (DPPH) assay to evaluate the antioxidative effects of OAK supplements. The assay procedure has been reported previously [20]. In brief, the OAK supplements were diluted in distilled water to prepare solutions with varying concentrations (0, 5, 10, 20, 50, and 100 μg/mL). Next, we added 100 μL of 1.5 mM/mL DPPH (D9132; Sigma-Aldrich Co., St. Louis, MO, USA) to a 96-well plate. Different concentrations of the OAK supplements were added to the DPPH-containing wells and incubated at room temperature for 30 min. Next, absorbance was measured at 517 nm by using a microplate spectrophotometer (µQuant, Biotek Instruments, Inc., Winooski, VT, USA). The DPPH assay was performed in triplicate for each OAK supplement concentration. To evaluate the antioxidative effects of OAK supplements, we measured the absorbance of various concentrations (0, 20, 50, 100, and 200 μg/mL) of blank methanol and L-ascorbic acid (control; A5960, Sigma-Aldrich). Blank methanol and L-ascorbic acid were mixed with DPPH and incubated at room temperature for 30 min before recording absorbance. The inhibitory activity (%) of OAK supplements in DPPH scavenging was calculated as follows: DPPH-scavenging activity of OAK supplements (%) = 100 × [(absorbance of OAK supplements + DPPH) − (absorbance of OAK blank)]/[(absorbance of DPPH) − (absorbance of methanol)].

### 2.3. Experimental Animal Preparation

Schematic experimental protocol for protective effect of oral OAK supplements on HFD-induced cardiovascular dysfunction in ApoE-knockout mice was shown in Figure 2. The experimental group comprised 24 male ApoE-knockout mice (aged 4 months), which were purchased from the National Laboratory Animal Center (NLAC; Taipei, Taiwan). By contrast, the wild-type (WT) control group comprised 12 C57/BL6 mice, which were purchased from the NLAC. The ApoE-knockout mice were divided into the normal diet (ND) group (ApoE^−/−^ ND; *n* = 6), HFD group (ApoE^−/−^/HFD; *n* = 6), HFD plus low-dose (LD) OAK supplements treatment group (LD, 150 mg/kg; ApoE^−/−^/HFD/LD; *n* = 6), and HFD plus high-dose (HD) OAK supplements treatment group (HD, 300 mg/kg; ApoE^−/−^/HFD/HD; *n* = 6). The control mice were divided into the ND group (WT/ND; *n* = 6) and the HFD group (WT/HFD; *n* = 6). HFD was fed ad libitum. The OAK supplements treatment group (ApoE^−/−^/HFD/LD and ApoE^−/−^/HFD/HD) was treated with OAK supplements twice a day. Changes in the body weight and blood pressure of mice were recorded every week. After 12 weeks of OAK supplements treatment, a cardiac ultrasonography was performed. Subsequently, all mice were euthanized. Their blood and cardiac and arterial tissues were collected. In addition, macrophages were isolated from their peritoneal cavity and cultured in vitro for subsequent experiments.

The protocol for animal experiments was approved by the Institutional Animal Care and Use Committee of National Taiwan Normal University (protocol number: 110017). All ApoE-knockout mice and C57/BL6 mice were housed in the animal facility of National Taiwan Normal University. The mice were raised in accordance with the international guidelines for the care and use of laboratory animals and bred in compliance with the relevant mouse breeding regulations. Our animal experiment design conforms to the 3R principles of “replacement”, “reduction”, and “optimization”.

### 2.4. Echocardiography

We assessed the cardiovascular performance of mice through color Doppler M-mode echocardiography. As described in our previous study [21], the experimental mice were placed on a heated work platform to monitor echocardiography and respiratory rate. Echocardiograms were measured using the Prospect high-resolution imaging system (S-Sharp Corporation, Taichung, Taiwan). Changes in heart rate, ejection fraction, cardiac output, isovolumic contraction time (IVCT), isovolumic relaxation time (IVRT), and ejection time (ET) were analyzed using M-mode and color Doppler images.

### 2.5. Isolation of Primary Murine Macrophage and Cell Culture

Primary mouse peritoneal macrophages were collected from each group of ApoE^−/−^ mice (*n* = 3) using a 10 mL syringe equipped with 25 G needle. The mice underwent intraperitoneal lavage injection with 10 mL of cold phosphate balanced solution (PBS, pH = 7.2~7.4), which was centrifuged at 1500 rpm for 10 min to pellet cells and incubated in Dulbecco’s modified Eagle medium (DMEM) with 10% fetal bovine growth serum (FBS) supplement and 2% bovine serum albumine (BSA); 1penicillin-streptomycin solution was obtained from 100 stock solution (ThermoFisher Scientific, Waltham, MA, USA). The primary peritoneal adherent cells (mostly macrophages) from each of the ApoE^−/−^ mice were adjusted to 2 × 10^6^ cells/mL in DMEM medium. Then, 2 × 10^5^ cells were seeded in 12-well coating plates per well and incubated at 37 °C in 5% CO_2_ humidified atmosphere for 48 h.

### 2.6. Hematological and Biochemical Analyses

At the end of the intervention, we anesthetized the mice and collected blood samples from their veins; for this, we used ethylenediaminetetraacetate-coated and uncoated vacuum blood collection tubes. Serum was separated through centrifugation at 750× *g* for 15 min and then stored in a −20 °C freezer until use. Hematocrit values and total counts of red blood cells (RBCs) and white blood cells (WBCs) were determined. To estimate the WBC count, blood smears were prepared and stained with Giemsa. Regarding blood biochemical analyses, the serum samples were assessed at Academia Sinica Taiwan Mouse Clinic for the following parameters: total cholesterol, LDL cholesterol, triglyceride, aspartate amino transferase (AST), and alanine amino transferase (ALT). The levels of the aforementioned parameters in the blood of the experimental and WT mice that had either received an HFD or had not were evaluated.

### 2.7. Oil Red O Staining

Oil Red O staining was performed to examine lipid deposition in the arterial vascular tissue of mice. As described in a relevant study [22], mouse aorta sections were fixed on glass slides with 2% paraformaldehyde and then washed with PBS. Then, the sections were stained with 5% Oil Red O at room temperature for 15 min. The stained sections were observed under an optical microscope (Olympus BH2 System Optical Microscope; Olympus Corporation, Tokyo, Japan).

### 2.8. Evaluation of ROS and Mitochondrial Membrane Potential

Through immunofluorescence staining, we evaluated mitochondrial membrane potential (MMP) and ROS levels in the macrophages isolated from the peritoneal cavity of the experimental and WT mice. As in our previous study [18], the ROS level was estimated using dichlorofluorescein diacetate (Molecular Probes, Eugene, OR, USA) as an oxidative fluorescent probe, whereas MMP was estimated using the lipophilic cationic fluorescent dye 3,3′-dihexyloxacarbocyanine iodide (Sigma-Aldrich). The mean fluorescent intensity was analyzed using a flow cytometer (FACS Calibur; BD Biosciences PharMingen, San Diego, CA, USA).

### 2.9. Immunofluorescence and Immunohistochemical Staining

Immunofluorescence staining was performed to evaluate the expression levels of vascular cell adhesion molecule (VCAM)-1, CD36, and ABCA1 receptors in the aortic tissue of the experimental and WT mice. Mouse aorta sections, including the aortic arch, were fixed on slides with 2% paraformaldehyde at room temperature. After washing with phosphate-buffered saline (PBS), the sections were incubated with VCAM-1 antibodies (Santa Cruz Biotechnology, Inc., Dallas, TX, USA), CD36 antibodies (Santa Cruz Biotechnology Inc.), and ABCA1 antibodies (Genetex, Irvine, CA, USA) diluted in 2% bovine serum albumin. After another wash with PBS, IgG-fluorescein isothiocyanate antibodies diluted in 1% bovine serum albumin were added to the sections. After a final wash with PBS, the nuclei were stained with 1 μg/mL diamino-2-benzindole (DAPI; D3286, Sigma-Aldrich). Fluorescence images were recorded using a Leica DM IRB Inverted Fluorescence Microscope (Leica Microsystems, Wetzlar, Germany). Fluorescence expression was analyzed using Leica Application Suite (version 4.12; Leica Microsystems).

Cardiac tissue specimens obtained from the experimental and WT mice were fixed with 4% formaldehyde (Sigma-Aldrich) and embedded in paraffin. The specimens were cut into 5 μm-thick sections by using a tissue microtome and then mounted on glass slides. Myocardial tissue sections were either stained with hematoxylin and eosin (Sigma-Aldrich; to examine tissue integrity) or stained immunohistochemically with superoxide dismutase (SOD)2 and tumor necrosis factor (TNF)-α antibodies (Cell Signaling Technology Inc., Danvers, MA, USA; 1 h at room temperature). Immunohistochemically stained tissue sections were incubated with biotinylated secondary antibodies (Novolink Polymer Detection System l; Leica Biosystems Newcastle Ltd., Newcastle, UK) for 30 min, and the avidin–biotin–horseradish peroxidase complex (Novolink Polymer Detection System l) for an additional 30 min. Immunostaining signals were visualized using the 3,3′-diaminobenzidine chromogen (Novolink Polymer Detection System 1). The slides were counterstained with hematoxylin (Novolink Polymer Detection System 1).

### 2.10. Western Blot

The experimental and WT mice were transcardially perfused with saline, and their cardiac tissue samples were collected and homogenized in buffer solution. Then, the cardiac proteins present in the isolated solution were quantified using a bicinchoninic protein assay kit (Thermo Fisher Scientific Inc.) and separated through polyacrylamide gel electrophoresis (Bionovas Pharmaceuticals Inc., Washington DC, USA). The separated proteins were transferred onto polyvinylidene fluoride membranes (GE Healthcare Life Sciences, Barrington, IL, USA). Antibodies against β-actin (Thermo Fisher Scientific Inc.), SOD2, and TNF-α (Cell Signaling Technology Inc.) were used. The primary antibodies were detected using appropriate horseradish peroxidase–conjugated secondary antibodies (Santa Cruz Biotechnology Inc.). Immunostained samples were visualized using the enhanced chemiluminescence substrate (Millipore) and quantified using ImageJ (version 1.48t; National Institutes of Health, Washington, DC, USA).

### 2.11. Statistical Analysis

Data are expressed in terms of the mean ± standard error of the mean (SEM) values. Differences between HFD-fed experimental and WT mice were evaluated using one-way or two-way analysis of variance. The Student–Newman–Keuls multiple comparison post hoc test was performed if the F value was significant. Significance was set at *p* < 0.05.

## 3. Results

### 3.1. DPPH-Scavenging Activity of OAK Supplements

Figure 3 depicts the effectiveness of OAK supplements in alleviating oxidative stress and scavenging free radicals. The results of the DPPH assay revealed that the 20, 50, and 100 μg/mL concentrations of OAK supplements exhibited stronger antioxidative effects and higher free radical-scavenging efficiency than did the other concentrations (Figure 3A). When comparing the free radical-scavenging efficiency of OAK supplements with that of standard L-ascorbic acid, we observed that the 20, 50, and 100 μg/mL concentrations of OAK supplements exhibited 84%, 82%, and 83% free radical-scavenging efficiency, respectively. These values were significantly higher than those obtained for the 5 and 10 μg/mL concentrations of OAK supplements (*p* < 0.01; Figure 3B).

### 3.2. Effects of OAK Supplements on Mouse Body Weight and Blood Pressure

Figure 4A shows the changes in the body weight of HFD-fed experimental and WT mice after LD-OAK and HD-OAK treatments. Experimental mice on HFD exhibited a significant trend of weight gain compared with WT mice fed on ND and HFD (*p* < 0.01, ApoE^−/−^/HFD vs. WT/ND, *p* < 0.05, ApoE^−/−^/HFD vs. WT/HFD; Figure 4A). Both LD-OAK and HD-OAK treatment significantly reduced the weight gain in ApoE KO mice fed on HFD. (*p* < 0.05, ApoE^−/−^/HFD/LD and ApoE^−/−^/HFD/HD vs. ApoE^−/−^/HFD; Figure 4A). 

Figure 3B shows the changes in the blood pressure of HFD-fed experimental and WT mice after LD-OAK and HD-OAK treatments. Experimental mice on HFD exhibited a significant trend of pressure compared with WT mice fed on ND (*p* < 0.01, ApoE^−/−^/HFD vs. WT/ND; Figure 4B) but not compared with WT mice fed on HFD (*p* > 0.05, ApoE^−/−^/HFD vs. WT/HFD; Figure 4B). Both LD-OAK or HD-OAK treatment significantly reduced the blood pressure in ApoE KO mice fed on HFD (*p* < 0.05, ApoE^−/−^/HFD/LD and ApoE^−/−^/HFD/HD vs. ApoE^−/−^/HFD; Figure 4A).

### 3.3. Effects of OAK Supplements on Hemocytosis and Serum Lipid Accumulation

Figure 5A depicts the blood smears of HFD- or ND-fed experimental mice subjected to LD-OAK or HD-OAK supplements treatment. Higher levels of RBC destruction and higher counts of WBC were noted in the experimental mice that did not receive OAK supplements treatment than in those that received LD-OAK or HD-OAK supplements treatment. Furthermore, RBC count was significantly lower in HFD-fed experimental mice than in ND-fed experimental mice (*p* < 0.01, ApoE^−/−^/HFD vs. ApoE^−/−^/ND; Figure 5Ab). Higher counts of RBC were noted in the HFD-fed experimental mice that received LD-OAK or HD-OAK supplements treatment than in those that did not receive OAK supplements treatment (*p* < 0.01, ApoE^−/−^/HFD/LD or ApoE^−/−^/HFD/HD vs. ApoE^−/−^/HFD; Figure 5Ab). We further observed that eosinophilic WBC count was abnormally increased in HFD-fed experimental mice but was obviously decreased in OAK supplements treatment groups. The reduction of eosinophilic WBC count was dose dependent on OAK treatment.

Figure 5B shows serum lipid accumulation in the HFD- or ND-fed experimental mice that received LD-OAK or HD-OAK supplements treatment. Higher levels of serum lipid accumulation were noted in HFD-fed experimental mice than in ND-fed experimental mice. Moreover, lower levels of fat accumulation were observed in the HFD-fed experimental mice that received LD-OAK or HD-OAK supplements treatment than in those that did not receive OAK supplements treatment (Figure 5B). 

### 3.4. Effects of OAK Supplements on Abnormal Blood Lipid Levels and Liver Damage

Figure 6 presents the total cholesterol, LDL, triglyceride, AST, and ALT levels in HFD- or ND-fed experimental mice subjected to LD-OAK or HD-OAK supplements treatment, as well as WT subjected to sham or HFD treatment. HFD-fed experimental mice exhibited significantly higher levels of total cholesterol (Figure 6A), LDL (Figure 6B), triglycerides (Figure 6C), AST (Figure 6D), and ALT (Figure 6E) than did ND- or HFD-fed WT mice (*p* < 0.01–0.05, ApoE^−/−^/HFD vs. WT/ND or WT/HFD; Figure 6). Furthermore, the HFD-fed experimental mice that received LD-OAK or HD-OAK supplements treatment exhibited significantly lower levels of total cholesterol (Figure 6A), LDL (Figure 6B), triglycerides (Figure 6C), AST (Figure 6D), and ALT (Figure 6E) than did those that did not receive OAK supplements treatment (*p* < 0.01, ApoE^−/−^/HFD/LD or ApoE^−/−^/HFD/HD vs. ApoE^−/−^/HFD; Figure 6).

### 3.5. Effects of OAK Supplements on Arterial Accumulation of Lipids and Abnormal Expression of VCAM-1, CD36, and ABCA1 Receptors

Figure 7A shows lipid accumulation in the arterial wall of HFD- or ND-fed experimental and WT mice subjected to LD-OAK or HD-OAK supplements treatment. Higher levels of lipid deposition were noted in the HFD-fed experimental mice than in HFD- or ND-fed WT mice. Moreover, lower levels of lipid accumulation were observed in the HFD-fed experimental mice that received LD-OAK or HD-OAK supplements treatment than in those that did not receive OAK supplements treatment. The findings of Oil Red O staining revealed high levels of lipid deposition in HFD-fed experimental mice; notably, the levels decreased after LD-OAK or HD-OAK supplements treatment (Figure 7B).

Figure 7C shows the expression of the VCAM-1 receptor in the arterial wall of HFD- or ND-fed experimental mice subjected to LD-OAK or HD-OAK supplements treatment. The findings of immunofluorescence staining revealed higher expression levels of the VCAM-1 receptor in HFD-fed experimental mice than in ND-fed experimental mice. However, the expression of VACM-1 was downregulated after LD-OAK or HD-OAK supplements treatment. The expression levels of VACM-1 were significantly lower in the HFD-fed experimental mice that received LD-OAK or HD-OAK supplements treatment than in those that did not receive OAK supplements treatment (*p* < 0.01, ApoE^−/−^/HFD/LD or ApoE^−/−^/HFD/HD vs. ApoE^−/−^/HFD; Figure 7Cb).

Figure 7D shows the expression of the scavenger CD36 receptor in the arterial wall of HFD- or ND-fed experimental mice subjected to LD-OAK or HD-OAK supplements treatment. Higher expression levels of CD36 receptor were observed in HFD-fed experimental mice than in ND-fed experimental mice. The expression of the CD36 receptor was downregulated after LD-OAK or HD-OAK supplements treatment. The expression levels of CD36 receptor were significantly lower in the HFD-fed experimental mice that received LD-OAK or HD-OAK supplements treatment than in those that did not receive OAK supplements treatment (*p* < 0.01, ApoE^−/−^/HFD/LD or ApoE^−/−^/HFD/HD vs. ApoE^−/−^/HFD; Figure 7Db).

Figure 7E shows the expression of the scavenger ABCA1 receptor in the aortic arch vessel wall of HFD- or ND-fed experimental mice subjected to LD-OAK or HD-OAK supplements treatment. Aortic arch is one of the earliest places to develop atherosclerosis on HFD. Lower expression levels of ABCA1 receptor were noted in HFD-fed experimental mice than in ND-fed experimental mice. The expression of ABCA1 receptor was upregulated after LD-OAK or HD-OAK supplements treatment. The expression levels of ABCA1 receptor were significantly higher in the HFD-fed experimental mice that received LD-OAK or HD-OAK supplements treatment than in those that did not receive OAK supplements treatment (*p* < 0.01, ApoE^−/−^/HFD/LD or ApoE^−/−^/HFD/HD vs. ApoE^−/−^/HFD; Figure 7Db).

### 3.6. Effects of OAK Supplements on Lipid Accumulation and Abnormal CD11b Expression, ROS Production, and MMP in Mouse Macrophages

Figure 8A shows lipid accumulation in macrophages isolated from the peritoneal cavity of HFD- or ND-fed experimental mice subjected to LD-OAK or HD-OAK supplements treatment. The Oil Red O staining revealed that lipid accumulation was almost invisible in macrophages isolated from ND-fed experimental mice but was prominent in those isolated from HFD-fed experimental mice. Furthermore, lipid accumulation was almost invisible in macrophages isolated from HFD-fed experimental mice subjected to LD-OAK or HD-OAK supplements treatment (Figure 8A).

Figure 8B–D presents the results of the immunofluorescence staining of CD11b, ROS, and MMP. CD11b expression and ROS production increased, but MMP decreased in macrophages isolated from HFD-fed experimental mice. By contrast, CD11b expression and ROS production decreased, but MMP increased in macrophages isolated from HFD-fed experimental mice subjected to LD-OAK or HD-OAK supplements treatment (Figure 8B–D).

### 3.7. Effects of OAK Supplements on Cardiac Dysfunction, Myocardial Oxidative Stress, and Inflammation

Figure 9A presents the cardiac ultrasound scans of HFD- or ND-fed experimental or WT mice subjected to LD-OAK or HD-OAK supplements treatment. HFD-fed experimental mice exhibited abnormal cardiac systolic function compared with the findings in their WT counterparts. However, the HFD-fed experimental mice that received LD-OAK or HD-OAK supplements treatment exhibited nearly normal cardiac systolic function compared with findings in those that did not receive OAK supplements treatment (Figure 9Aa). Furthermore, HFD-fed experimental mice exhibited significantly prolonged IVCT, IVRT, and ET compared with the findings in their WT counterparts fed with or without an HFD (*p* < 0.01, ApoE^−/−^/HFD vs. ApoE^−/−^/ND or WT/ND; Figure 9Ab). The HFD-fed experimental mice that received LD-OAK or HD-OAK supplements treatment exhibited significantly reduced IVCT, IVRT, and ET compared with the findings in those that did not receive OAK supplements treatment (*p* < 0.01, ApoE^−/−^/HFD/LD or ApoE^−/−^/HFD/HD vs. ApoE^−/−^/HFD; Figure 9Ab).

Figure 9B shows myocardial SOD2 expression in HFD- or ND-fed experimental and WT mice subjected to LD-OAK or HD-OAK supplements treatment. Immunohistochemical staining revealed upregulated expression myocardial TNF-α in HFD-fed experimental mice; however, the expression of TNF-α was downregulated after LD-OAK or HD-OAK supplements treatment (Figure 9Ba). The expression levels of TNF-α were significantly higher in HFD-fed experimental mice than in experimental or WT mice not fed with an HFD (*p* < 0.01, ApoE^−/−^/HFD vs. ApoE^−/−^/ND or WT/ND; Figure 9Bb). The expression levels of SOD2 were significantly higher in the HFD-fed experimental mice that received LD-OAK or HD-OAK supplements treatment than in those that did not receive OAK supplements treatment (*p* < 0.01, ApoE^−/−^/HFD/LD or ApoE^−/−^/HFD/HD vs. ApoE^−/−^/HFD; Figure 9Bb).

Figure 9C shows myocardial TNF-α expression in HFD- or ND-fed experimental and WT mice subjected to LD-OAK or HD-OAK supplements treatment. Immunohistochemical staining indicated a downregulated expression of myocardial SOD2 in HFD-fed experimental mice. The expression of TNF-α was upregulated after LD-OAK or HD-OAK supplements treatment (Figure 9Ca). Western blotting indicated significantly higher expression levels of TNF-α in HFD-fed experimental mice than in experimental or WT mice not fed with an HFD (*p* < 0.01, ApoE^−/−^/HFD vs. ApoE^−/−^/ND or WT/ND; Figure 9Cb). The expression levels of TNF-α were significantly lower in the HFD-fed experimental mice that received LD-OAK or HD-OAK supplements treatment than in those that did not receive OAK supplements treatment (*p* < 0.01, ApoE^−/−^/HFD/LD or ApoE^−/−^/HFD/HD vs. ApoE^−/−^/HFD; Figure 9Cb).

## 4. Discussion

Our recent findings provided insights into the effectiveness of antcin K in treating atherosclerosis [23]. Our laboratory experiments revealed that the combined use of ovatodiolide and antcin K was more effective in treating atherosclerosis than their individual effectiveness. This prompted us to evaluate the therapeutic effects of OAK supplements on HFD-induced cardiovascular dysfunction in ApoE-knockout mice. OAK supplement concentrations of 20, 50, and 100 μg/mL exhibited stronger antioxidative effects and higher (<80%) free radical-scavenging efficiency than did the other concentrations (Figure 3A).

In this study, HFD-fed ApoE-knockout mice were used to determine the therapeutic effects of OAK supplements on cardiovascular dysfunction because human studies require a long follow-up period; cardiovascular dysfunction in humans occurs over a period ranging from months to years or even decades. When fed with an HFD or a high-cholesterol diet, ApoE-knockout mice can develop atherosclerotic lesions; several features of cardiovascular disease are similar between these mice and humans [24]. After feeding the mice with an HFD for 12 weeks, we noted that mice began to exhibit symptoms of cardiovascular abnormalities. Compared with WT control mice, HFD-fed ApoE-knockout mice had increased body weight, blood pressure, WBC count, blood lipid accumulation, total cholesterol level, LDL level, AST level, and ALT level, but a reduced RBC count. After oral treatment with OAK supplements twice a day for 12 weeks, HFD-fed ApoE-knockout mice had nearly normal body weight and blood pressure (Figure 4); normal RBC and WBC counts; nearly clear plasma (Figure 5); and normal total cholesterol, LDL, triglyceride, AST, and ALT levels (Figure 6). These findings indicate the therapeutic effects of OAK supplements on cardiovascular dysfunction.

HFD-fed experimental mice exhibited increased lipid deposition and upregulated VCAM-1 and CD36 receptor expression but downregulated ABCA1 receptor expression (Figure 7). After oral treatment with OAK supplements, the mice exhibited reduced lipid deposition and downregulated VCAM-1 and CD36 receptor expression but upregulated ABCA1 receptor expression (Figure 7). 

During the development of atherosclerosis, the VCAM-1 receptor facilitates monocyte migration and inflammation and promotes macrophage adhesion and migration [25]. The production of TNF-α and interleukin (IL)-1β enhances the expression of the VCAM-1 receptor in vascular endothelial cells [26]. Some scavenger receptors on endothelial macrophages, particularly the CD36 receptor, can prompt macrophages to phagocytize oxidized LDLs, which accelerate the accumulation of cholesterol in vascular endothelial cells [27]. The ABCA1 receptor can help maintain plasma levels of high-density lipoprotein, thus reducing the risk of HFD-induced atherosclerosis [28].

Our results revealed increased lipid accumulation, CD11b expression, and ROS production but reduced MMP in macrophages isolated from HFD-fed ApoE-knockout mice (Figure 8). After oral treatment with OAK supplements, we noted reduced lipid accumulation, CD11b expression, and ROS production but increased MMP in macrophages isolated from these mice (Figure 8). During the development of atherosclerosis in HFD-fed ApoE-knockout mice, monocytes can migrate to the subendothelial space of the intima; this process is mediated by VCAM-1. Subsequently, the migrated monocytes differentiate into macrophages, which pass through the intercellular space and remove oxidized LDLs [29,30]. The uptake of cholesterol by macrophages in atherosclerotic plaques may induce the generation of ROS in the cytoplasm, which, in turn, leads to the secretion of the inflammatory cytokines TNF-α and IL-1β. In addition, endothelial injury caused by HFD-induced atherosclerosis may enhance the production of ROS in vascular endothelial cells, thereby promoting the secretion of the inflammatory cytokines TNF-α and IL-1β [31]. ROS production in response to LDL oxidation is regulated by macrophages [32,33].

HFD-fed ApoE-knockout mice with cardiac dysfunction exhibited abnormal IVCT, IVRT, and ET and a downregulated expression of the antioxidation-related protein SOD2; however, they exhibited an upregulated expression of the inflammation-related protein TNF-α in their myocardial tissue (Figure 9). After oral treatment with OAK supplements, the markers of cardiac dysfunction in HFD-fed ApoE-knockout mice returned to the corresponding near-normal levels. Moreover, the expression of SOD2 was upregulated, while that of TNF-α was downregulated in the myocardial tissue of HFD-fed ApoE-knockout mice (Figure 9). HFD-induced TNF-α secretion in the myocardial tissue can stimulate the adhesion of monocytes to endothelial cells and cause atherosclerosis [34]. A study reported that an HFD induced hypertrophy of the cardiac tissue in mice, resulting in increased force and time of myocardial contraction [35]. Moreover, HFD-fed mice exhibited increased ROS levels in the cardiac tissue; an increased ROS level may cause myocardial damage [36]. Our results revealed that an HFD induced cardiac dysfunction in ApoE-knockout mice through ROS generation and mitochondrial oxidation in the cardiac tissue, and OAK supplements treatment protected cardiac function by reducing oxidative stress.

## 5. Conclusions

Our findings indicate that ApoE-knockout mice developed cardiovascular dysfunction after being fed with an HFD for 12 weeks. These mice exhibited abnormal body weight gain; elevated blood pressure; reduced RBC but increased WBC counts; and increased levels of total cholesterol, LDL, triglycerides, AST, ALT, and lipid accumulation in the blood. We noted increased levels of lipid accumulation and an upregulated expression of VCAM-1 and CD36 receptors but a downregulated expression of ABCA1 receptor in the arterial wall of HFD-fed ApoE-knockout mice. In addition, we observed lipid accumulation, increased ROS production, upregulated CD11b expression, and reduced MMP in macrophages isolated from the peritoneal cavity of mice. HFD-fed experimental mice exhibited abnormal IVCT, IVRT, and ET and reduced SOD2 expression but increased TNF-α expression. Oral OAK supplements treatment mitigated cardiovascular dysfunction in HFD-fed ApoE-knockout mice, as indicated by their significantly reduced body weight and blood pressure; restored normal counts of RBC and WBC; and normal levels of total cholesterol, LDL cholesterol, triglycerides, AST, and ALT. Moreover, after OAK supplements treatment, reduced levels of lipid deposition and a downregulated expression of VCAM-1 and CD36 receptors but upregulated expression of ABCA1 receptor were noted in the arterial wall of HFD-fed ApoE-knockout mice. These findings suggest that oral OAK supplements treatment can mitigate HFD-induced cardiovascular dysfunction by reducing lipid accumulation in the arteries and oxidative stress and inflammation in the cardiovascular tissue.

## Figures and Tables

**Figure 1 nutrients-15-04074-f001:**
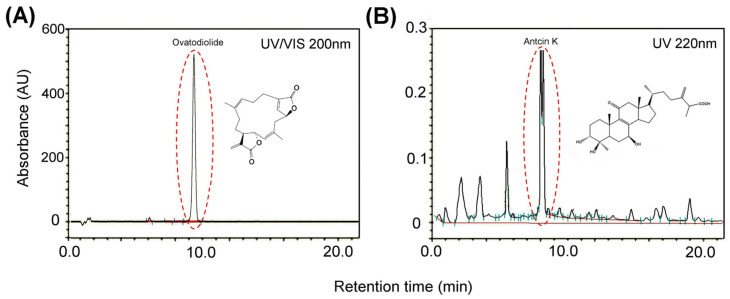
Chromatographic fingerprint of ovatodiolide and antcin K. (**A**) Ovatodiolide was isolated from *Anisomeles indica*, and (**B**) antcin K was isolated from *Antrodia camphorata*.

**Figure 2 nutrients-15-04074-f002:**
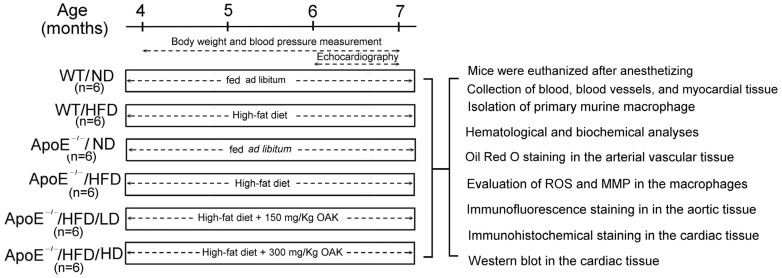
Schematic experimental protocol for investigating the protective effect of OAK supplements on HFD-induced cardiovascular dysfunction in ApoE-knockout mice. ROS: reactive oxygen species; MMP: mitochondrial membrane potential.

**Figure 3 nutrients-15-04074-f003:**
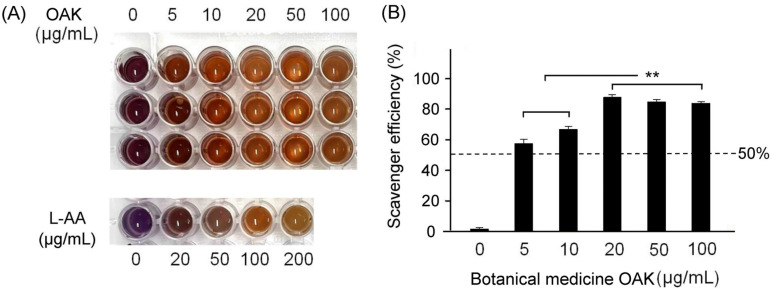
Antioxidative effects of OAK supplements. (**A**) DPPH assay was performed using various OAK supplements concentrations. L-ascorbic acid was used as a standard antioxidative compound. (**B**) Comparison of free radical-scavenging efficiency among various concentrations of OAK supplements (A0, 5, 10, 20, 50, and 100 μg/mL; *n* = 3 for each group). Data are presented in terms of the mean ± SEM values. ** *p* < 0.01, one-way ANOVA followed by the Student–Newman–Keuls multiple comparison post hoc test.

**Figure 4 nutrients-15-04074-f004:**
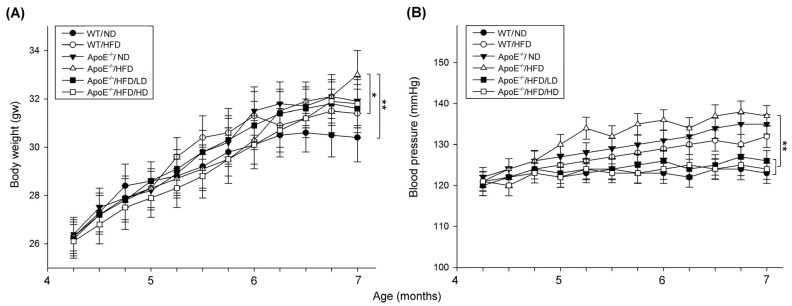
Effects of OAK supplements on the body weight and blood pressure of HFD-fed ApoE-knockout mice and their WT counterparts. (**A**) Comparison of body weight between ApoE-knockout mice and their WT counterparts fed with a high-fat diet (HFD) or a normal diet (ND) and subjected to low-dose (LD)-OAK or high-dose (HD)-OAK supplements treatment. (**B**) Comparison of blood pressure between ApoE-knockout mice and their WT counterparts fed with an HFD or an ND and subjected to LD-OAK or HD-OAK supplements treatment (*n* = 6 for each group). Data are presented in terms of the mean ± SEM values. ** *p* < 0.01 and * *p* < 0.05, two-way ANOVA followed by the Student–Newman–Keuls multiple comparison post hoc test. WT/ND: wild-type mice fed with an ND, WT/HFD: wild-type mice fed with an HFD, ApoE^−/−^/HFD: ApoE-knockout mice fed with an HFD, ApoE^−/−^/HFD/LD: ApoE-knockout mice fed with an HFD and treated with LD-OAK supplements, ApoE^−/−^/HFD/HD: ApoE-knockout mice fed with an HFD and treated with HD-OAK supplements.

**Figure 5 nutrients-15-04074-f005:**
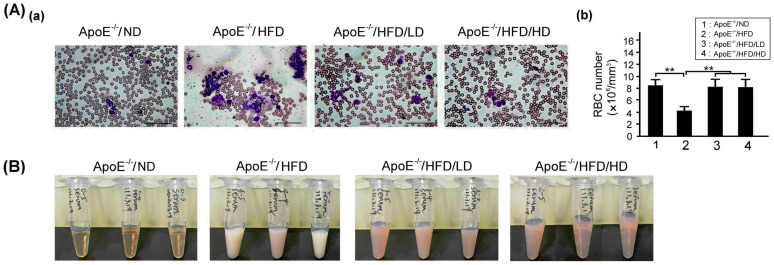
Effects of OAK supplements on the blood cell counts of HFD-fed ApoE-knockout mice. (**A**) (**a**) Blood smears of HFD- or ND-fed ApoE-knockout mice subjected to LD-OAK or HD-OAK supplements treatment. Scale bar = 100 μm. (**b**) Comparison of the RBC count among ApoE-knockout mice fed with an ND or an HFD and subjected to LD-OAK or HD-OAK supplements treatment (*n* = 6 for each group). Data are presented in terms of the mean ± SEM values. ** *p* < 0.01, one-way ANOVA followed by the Student–Newman–Keuls multiple comparison post hoc test. (**B**) Lipid accumulation in the serum of ApoE-knockout mice fed with an ND or HFD and subjected to LD-OAK or HD-OAK supplements treatment.

**Figure 6 nutrients-15-04074-f006:**
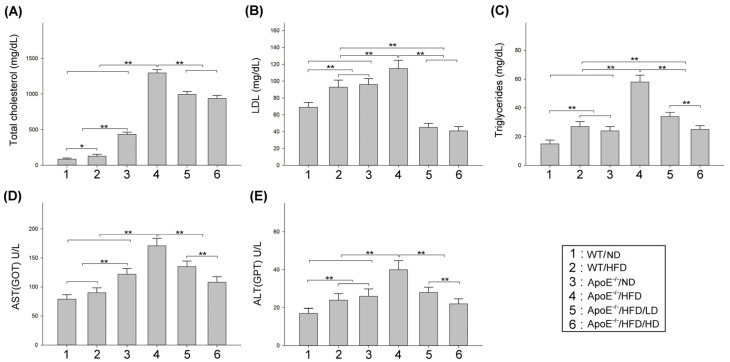
Effects of OAK supplements on abnormal blood lipid levels and liver damage in HFD-fed ApoE-knockout mice. Comparison of the levels of (**A**) total cholesterol, (**B**) low-density lipoprotein (LDL), (**C**) triglycerides, (**D**) aspartate amino transferase (AST), and (**E**) alanine amino transferase (ALT) among ApoE-knockout mice fed with an ND or an HFD and subjected to LD-OAK or HD-OAK supplements treatment (*n* = 6 for each group). Data are presented in terms of the mean ± SEM values. ** *p* < 0.01 and * *p* < 0.05, one-way ANOVA followed by the Student–Newman–Keuls multiple comparison post hoc test.

**Figure 7 nutrients-15-04074-f007:**
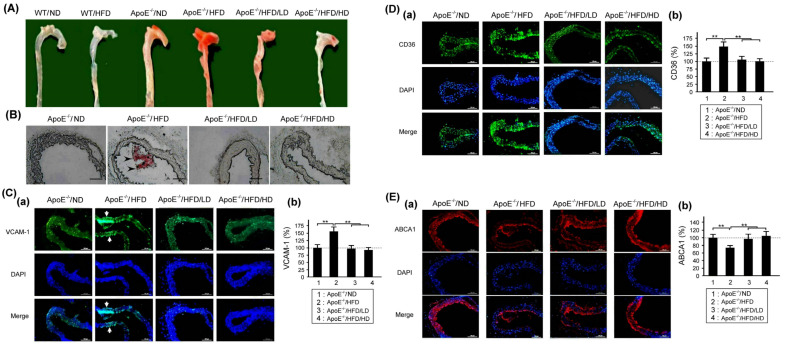
Effects of OAK supplements on the arterial accumulation of lipids and expression of VCAM-1, CD36, and ABCA1 receptors in the aortic arch vessel wall of HFD-fed ApoE-knockout mice. (**A**) Lipid accumulation. (**B**) Lipid accumulation assessed through Oil Red O staining. (**C**–**E**) Representative immunofluorescence staining of VCAM-1 (**Ca**), CD36 (**Da**), and ABCA1 (**Ea**); (**b**) Comparison of the immunofluorescence expressions of VCAM-1 (**Cb**), CD36 (**Db**), and ABCA1 (**Eb**) among ApoE-knockout mice fed with an ND or an HFD and subjected to LD-OAK or HD-OAK supplements treatment (*n* = 6 for each group). (**B**–**E**) The arrow indicates the location of lipid accumulation. Scale bar = 100 μm. Data are presented in terms of the mean ± SEM values. ** *p* < 0.01, one-way ANOVA followed by the Student–Newman–Keuls multiple comparison post hoc test.

**Figure 8 nutrients-15-04074-f008:**
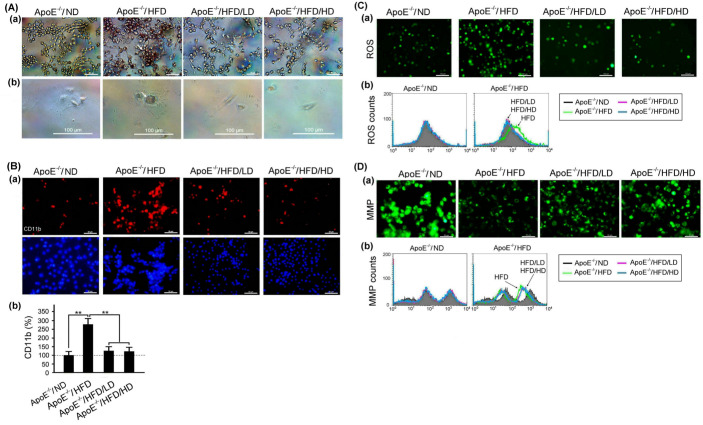
Effects of OAK supplements on lipid accumulation and abnormal CD11b expression, ROS production, and MMP in macrophages isolated from HFD-fed ApoE-knockout mice. (**A**) (**a**) Lipid accumulation (assessed using Oil Red O staining) in macrophages isolated from HFD- or ND-fed ApoE-knockout mice subjected to LD-OAK or HD-OAK supplements treatment. Scale bar = 50 μm. (**b**) Enlarged views of lipid accumulation in macrophages isolated from HFD- or ND-fed ApoE-knockout mice subjected to LD-OAK or HD-OAK supplements treatment. Scale bar = 100 μm. (**B**–**D**) CD11b expression (**Ba**), ROS production (**Ca**), and MMP (**Da**) in macrophages isolated from HFD- or ND-fed ApoE-knockout mice subjected to LD-OAK or HD-OAK supplements treatment (assessed through immunofluorescence staining). Scale bar = 50 μm. Comparison of CD11b expression (**Bb**), ROS production (**Cb**), and MMP (**Db**) among macrophages shown in isolated from HFD- or ND-fed ApoE-knockout mice subjected to LD-OAK or HD-OAK supplements treatment. (*n* = 6 for each group). Data are presented in terms of the mean ± SEM values. ** *p* < 0.01, one-way ANOVA followed by the Student–Newman–Keuls multiple comparison post hoc test.

**Figure 9 nutrients-15-04074-f009:**
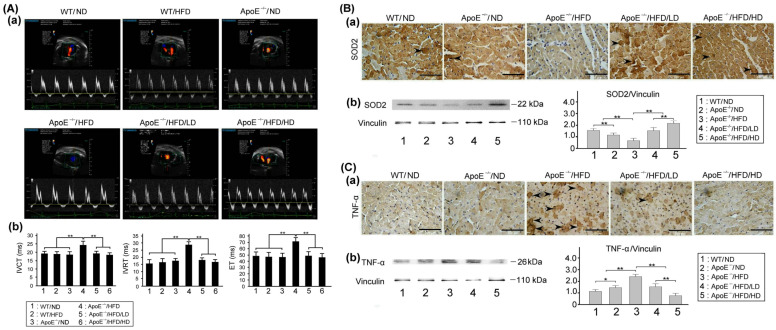
Effects of OAK supplements on cardiac dysfunction, myocardial oxidative stress, and inflammation in HFD-fed ApoE-knockout mice. (**A**) Cardiac ultrasound scan of HFD- or ND-fed ApoE-knockout mice and their WT counterparts subjected to LD-OAK or HD-OAK supplements treatment (**Aa**). Comparison of the isovolumic contraction time (IVCT), isovolumic relaxation time (IVRT), and ejection time (ET) among ApoE-knockout mice and their WT counterparts fed with an ND or an HFD and subjected to LD-OAK or HD-OAK supplements treatment (**Ab**). (**B**,**C**) Cardiac expression levels of SOD2 (**Ba**) and TNF-α (**Ca**) in HFD- or ND-fed ApoE-knockout and WT mice subjected to LD-OAK or HD-OAK supplements treatment. Comparison of expressions of SOD2 (**Bb**) and TNF-α (**Cb**) in HFD- or ND-fed ApoE-knockout and WT mice subjected to LD-OAK or HD-OAK supplements treatment. The arrow indicates the expressions of SOD2 (**Ba**) and TNF-α (**Ca**). Scale bar = 200 μm. (*n* = 6 for each group). Data are presented in terms of the mean ± SEM values. ** *p* < 0.01 and * *p* < 0.05, one-way ANOVA followed by the Student–Newman–Keuls multiple comparison post hoc test.

## Data Availability

The datas are confidential.

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
