# Peer review of "Alleviating Effects of Ovatodiolide and Antcin K Supplements on High-Fat Diet-Induced Cardiovascular Dysfunction in ApoE-Knockout Mice by Attenuating Oxidative Stress"

_nutrients, 2023, doi:10.3390/nu15184074_

Round 1

Reviewer 1 Report

The authors evaluate the therapeutic effects of OAK (two different compounds with similar effects) supplements on HFD-induced cardiovascular dysfunction in ApoE-knockout mice. The study was well-designed and conducted. The results are well presented. 

Why did the authors not test the two compounds separately?

Include a further discussion on the effects of Ovatodiolide and Antcin K and how they can be more effective in their combined use. 

Minor: 

Change the word sacrifice to euthanize because this word may have a religious connotation. 

Author Response

Response to Reviewer 1 Comments

Dear Reviewer 1,

Many thanks for valuable comments. In accordance with the concerns you have identified, point-for-point responses to your comments and questions are given below by using red description in our review reports. Also, we have corrected incorrect use in the text and let our manuscript been reviewed and proofreaded by Wallace Academic Editing. Please check them in the revised manuscript. If there is any problem, please address all correspondence concerning the manuscript to Dr. Chung-Hsin Wu.

Reviewer 1 comments

Comments and Suggestions for Authors

Point 1: The authors evaluate the therapeutic effects of OAK (two different compounds with similar effects) supplements on HFD-induced cardiovascular dysfunction in ApoE-knockout mice. The study was well-designed and conducted. The results are well presented.

Why did the authors not test the two compounds separately?

Include a further discussion on the effects of Ovatodiolide and Antcin K and how they can be more effective in their combined use.

Response Point 1: Many thanks the Reviewer's comments. We have added explanations about how Ovatodiolide and Antcin K can be more effective in their combined usein the text as follows: "Our recent findings provided insights into the effectiveness of antcin K in treating atherosclerosis [23, Lu et al., 2023]. Our laboratory experiments revealed that the combined use of ovatodiolide and antcin K was more effective in treating atherosclerosis than their individual effectiveness. This prompted us to evaluate the therapeutic effects of OAK sup-plements on HFD-induced cardiovascular dysfunction in ApoE-knockout mice." Please check them in the revised manuscript.

Point 2: Minor:

Change the word sacrifice to euthanize because this word may have a religious connotation.

Response Point 2: Many thanks the Reviewer's comments. We have changed the word sacrifice to euthanize in the text. Please check them in the revised manuscript.

Reviewer 2 Report

The work from Lu and colleagues investigated the effects of the combination of ovatodiolide and antcin K (OAK) supplements on high fat diet induced cardiovascular dysfunction in ApoE knockout mice. Overall the work is well designed and presented, but I have the following comments:

1.       On page 7:

Row 260-261: should not only compare experimental mice on HFD to WT on ND, maybe change to Experimental mice on HFD exhibited a significant trend of weight gain compared with WT mice on ND and HFD.

Row 262-264: Both LD-OAK or HD-OAK treatment significantly reduced the weight gain in ApoE KO mice fed on HFD.

Figure 4: There are two control groups with HFD or ND, author only compared ApoE KO with HFD to WT on ND, which should be clarified clearly.

2.       On page 8:

Figure 5: Seems like there are a lot of wbc in ApoE HFD group, and decreased in OAK treatment groups from the staining, did you check the components of wbc such as lymphocyte, neutrophils and monocytes, also is it dose depended?

Row 314: WT are not subjected to treatment, please clearly describe

3.       On page 10:

Figure 7:  (A) do you include aortic arch as it is the earliest places to develop atherosclerosis on HFD, WT/HFD should have  some lesions in aortic arch after 12 weeks HFD. (B). Which part of aorta do you do the staining?

It is better to get native speaker for language modification

Author Response

Response to Reviewer 2 Comments

Dear Reviewer 2,

Many thanks for valuable comments. In accordance with the concerns you have identified, point-for-point responses to your comments and questions are given below by using red description in our review reports. Also, we have corrected incorrect use in the text and let our manuscript been reviewed and proofreaded by Wallace Academic Editing. Please check them in the revised manuscript. If there is any problem, please address all correspondence concerning the manuscript to Dr. Chung-Hsin Wu.

Reviewer 2 comments

Comments and Suggestions for Authors

The work from Lu and colleagues investigated the effects of the combination of ovatodiolide and antcin K (OAK) supplements on high fat diet induced cardiovascular dysfunction in ApoE knockout mice. Overall the work is well designed and presented, but I have the following comments:

Point 1: On page 7:

Row 260-261: should not only compare experimental mice on HFD to WT on ND, maybe change to Experimental mice on HFD exhibited a significant trend of weight gain compared with WT mice on ND and HFD.

Row 262-264: Both LD-OAK or HD-OAK treatment significantly reduced the weight gain in ApoE KO mice fed on HFD.

Figure 4: There are two control groups with HFD or ND, author only compared ApoE KO with HFD to WT on ND, which should be clarified clearly.

Response Point 1: Many thanks the Reviewer's comments. We have compared experimental mice on HFD to WT on ND, and changed to "Experimental mice on HFD exhibited a significant trend of weight gain compared with WT mice on ND and HFD" in Row 260-261, and "Both LD-OAK or HD-OAK treatment significantly reduced the weight gain in ApoE KO mice fed on HFD" in Row 262-264 based on reviewer suggestions. Please check them in the revised manuscript.

Point 2: On page 8:

Figure 5: Seems like there are a lot of wbc in ApoE HFD group, and decreased in OAK treatment groups from the staining, did you check the components of wbc such as lymphocyte, neutrophils and monocytes, also is it dose depended?

Row 314: WT are not subjected to treatment, please clearly describe

Response Point 2: Many thanks the Reviewer's comments. We have added explanations in the text as follows: "After checking the components of WBC in ApoE HFD group, we observed that eosinophilic WBC count was abnormally increased ApoE HFD group, but decreased in OAK treatment groups from the staining. Reduction of eosinophilic WBC count was dose depended on OAK treatment." Also, WT in Row 314 was clearly describe. Please check them in the revised manuscript.

Point 3: On page 10:

Figure 7:  (A) do you include aortic arch as it is the earliest places to develop atherosclerosis on HFD, WT/HFD should have some lesions in aortic arch after 12 weeks HFD. (B). Which part of aorta do you do the staining?

Response Point 3: Many thanks the Reviewer's comments. As suggested by the reviewer, we mainly perform IHC staining on the aortic arch vessel wall in Figure 7. We have added explanations in the text. Please check them in the revised manuscript.

Point 4: Comments on the Quality of English Language

It is better to get native speaker for language modification

Response Point 4: Many thanks the Reviewer's comments. The manuscript has been reviewed and proofreaded by Wallace Academic Editing. Please check them in the revised manuscript.